# A Strategy for the Production and Molecular Validation of Agrobacterium-Mediated Intragenic Octoploid Strawberry

**DOI:** 10.3390/plants10112229

**Published:** 2021-10-20

**Authors:** Ke Duan, Ying-Jie Zhao, Zi-Yi Li, Xiao-Hua Zou, Jing Yang, Cheng-Lin Guo, Si-Yu Chen, Xiu-Rong Yang, Qing-Hua Gao

**Affiliations:** 1Shanghai Key Laboratory of Protected Horticultural Technology, Forestry and Fruit Tree Research Institute, Shanghai Academy of Agricultural Sciences (SAAS), Shanghai 201403, China; zouxh_113@126.com (X.-H.Z.); 20170304@saas.sh.cn (J.Y.); yxr000120@163.com (X.-R.Y.); 2Lanzhou New Area Academy of Modern Agricultural Sciences, Lanzhou 730300, China; yingjie428921@163.com; 3Ecological Technique and Engineering College, Shanghai Institute of Technology, Shanghai 201418, China; lzy18185768772@163.com; 4Hangzhou Woosen Biotechnology Co., Ltd., Hangzhou 310012, China; guochenglin@ws-bio.com; 5College of Food Science, Shanghai Ocean University, Shanghai 201306, China; csy13023156961@163.com

**Keywords:** *Fragaria* × *ananassa*, sugar transporter, intragenesis, *Agrobacterium*-mediated transformation, droplet digital PCR

## Abstract

Intragenesis is an all-native engineering technology for crop improvement. Using an intragenic strategy to bring genes from wild species to cultivated strawberry could expand the genetic variability. A robust regeneration protocol was developed for the strawberry cv. ‘Shanghai Angel’ by optimizing the dose of Thidiazuron and identifying the most suitable explants. The expression cassette was assembled with all DNA fragments from *F. vesca,* harboring a sugar transporter gene *FvSTP8* driven by a fruit-specific *FvKnox* promoter. Transformed strawberry was developed through an *Agrobacterium*-mediated strategy without any selectable markers. Other than PCR selection, probe-based duplex droplet digital PCR (ddPCR) was performed to determine the T-DNA insert. Four independent transformed shoots were obtained with a maximum of 5.3% efficiency. Two lines were confirmed to be chimeras, while the other two were complete transformants with six and 11 copies of the intragene, respectively. The presence of a vector backbone beyond the T-DNA in these transformants indicated that intragenic strawberries were not obtained. The current work optimized the procedures for producing transformed strawberry without antibiotic selection, and accurately determined the insertion copies by ddPCR in the strawberry genome for the first time. These strategies might be promising for the engineering of ‘Shanghai Angel’ and other cultivars to improve agronomic traits.

## 1. Introduction

Cultivated strawberry (*Fragaria × ananassa*) is planted worldwide due to its high nutritional and economic value. Recently, the origin and evolution of cultivated strawberry have been elucidated, confirming that its single dominant subgenome provider is the diploid woodland strawberry *F. vesca* [1]. An improved genome and high-quality annotations of *F. vesca* [2,3], together with the updated annotations of the *F. × ananassa* genome [4], built an integrated science context opening a new era of gene functional studies and genetic engineering in cultivated strawberry. The key areas include regulating strawberry flowering, fruit development and quality, and disease resistance [5]. The unique domestication and evolution history limited the genetic variability in modern cultivated strawberry. This fact hinders its improvement with conventional breeding and constitutes a strong motivation to use engineering and gene editing to improve strawberry, as we could learn from *Solanaceae* crops [6].

Molecular breeding technologies have addressed some constraints on conventional breeding for enhanced resistance, flavor, and nutritional quality of crops [7]. Until 2020, only five genetically engineered fruit crops had been approved for commercial purposes, but more than 45 novel fruit germplasms belonging to 20 distinct crops had been successfully developed with accelerated molecular techniques [8]. Many efforts utilizing genetic engineering methods have improved the important agronomic traits of strawberry [9]. For example, fruit-specific antisense suppression of the ADP glucose pyrophosphorylase catalyzing starch biosynthesis increased the soluble sugar content and decreased starch accumulation in ripe fruits of the strawberry cv. Anther [10]. Overexpression of the transcriptional activator *FaMYB10* elevated anthocyanin expression in many organs beyond those of strawberry fruits [11]. Fruit-specific antisense suppression of the endo-b-1,4-glucanase (EGase) catalyzing the disassembly of hemicellulose molecules in the cell wall increased strawberry fruits’ firmness [12]. Overexpression of the endogenous polygalacturonase-inhibiting protein gene *FaPGIP* driven by the promoter of the fruit-specific gene *FaExp2* was achieved to regulate strawberry’s resistance to the fruit rot fungus *Botrytis cinerea* [13]. Overexpression and CRISPR/Cas9-directed mutagenesis of RAP, a glutathione S transferase (GST) (a carrier transferring anthocyanins from the cytosol into the vacuole) reversed the coloration of strawberry fruits [14].

Due to the low frequency of plant transformation, markers, including selectable types, such as hygromycin and kanamycin antibiotic resistance, and screenable types, such as visional GFP and GUS, have long been used to identify rare transformants [15]. However, the use of markers provokes public concerns about the biosafety of engineered crops. Many researchers have endeavored to generate marker-free engineered plants via omitting markers in the transformation or eliminating markers after the transformation. Robust *Agrobacterium*-mediated transformation of potato [16,17,18] and wheat [19] enabled the production of marker-free transformants of these crops without selection. Marker-removal systems, such as the chemically inducible recombinase system and the bifunctional selectable system (the former for positive selection for transformants and the latter for negative selection for marker deletion), facilitated the development of the marker-free strawberry cv. ‘Calypso’ [20]. The same vector and marker-removal strategy was later applied to generate marker-free apple [21] and marker-free banana [22]. In a model tobacco plant, a double T-DNA vector was changed to a head-to-head pattern for the RB and LB directions, which significantly increased the frequency of marker-free plants in a co-transformation system [23].

To reduce misleading conceptual assumptions about genetically modified organisms (GMOs) and encourage focus on products rather than on the biotechnology and process, intragenic and transgenic categories were proposed [24]. A more precise and explicit nomenclature—‘intragenesis’—was defined as rearrangements of genomic material from the same sexual compatibility group and a lack of foreign DNA [25]. This biotechnological concept was further explored in genetic engineering as a novel all-native gene technology using natural DNA fragments in a new combination to produce genetic diversity and novel phenotypes [26,27,28,29,30,31,32,33]. Intragenesis was gradually accepted as an innovative tool for crop improvement in precision breeding since it produced fewer biosafety concerns and has higher consumer acceptance than transgenesis. The intragenic approach was useful in bringing back lost traits and genetic diversity to the crop from wild germplasms [34]. For fruit crop improvement, intragenesis was recommended as the ideal approach [35]. Many intragenic vectors have been developed and tested [36]. Intragenic apple with enhanced resistance to scrab was developed [37] and the transformation systems for intragenic apple were further optimized [38]. Intragenic potato exhibiting resistance to bruising and discoloration has been granted commercial approval [7,39]. Intragenic black carrot has been proposed [40]. Intragenic strawberry was also developed, although the resistance trait had not been improved as expected [13].

A central task in genetic engineering is to validate the transformation and identify the copy number of the DNA insertion. Droplet digital PCR (ddPCR) was developed to detect DNA by partitioning individual amplifications into separate droplets and detecting the endpoint amplification products [41]. This method has been demonstrated to be more sensitive and have a lower error rate than qPCR and has been successfully used to determine the transgenic copy number in banana [22], tobacco [42], and maize [43]. A comparative study using ddPCR and Southern blotting in an array of genetically modified crops established single duplex ddPCR as an accurate method for measuring transgenic insertion events [44]. DdPCR has become a simple and cost-efficient method for analyzing copy number variations and is widely used in biology [45].

So far, use of the novel intragenesis tool has scarcely been reported in strawberry except to increase fruit resistance to *Botrytis cinerea* [13]. In addition, although the *Agrobacterium*-mediated transformation of strawberry is easy to perform, it is challenging to use DNA blotting in determining the transgenic copy number in the octoploid genome. In genetic engineering, plants with a single copy insertion of new DNA are desired for stable transferability to progeny. To meet these gaps, here we assembled a marker-free vector harboring an intragene-expressing cassette with DNA fragments from *F. vesca.* Then, relying on a highly efficient regeneration method, genetically transformed strawberry plantlets were developed using *Agrobacterium*-mediated transformation without selection. Following direct PCR selection, the T-DNA insertion copy number in these candidate transformants was further determined by a single duplex ddPCR. To our knowledge, this is the first report of an accurate analysis of copy number by ddPCR in genetically modified strawberry.

## 2. Results

### 2.1. Optimization of Shoot Regeneration from Leaf Explants of the Strawberry cv. ‘Shanghai Angel’

During in vitro culture, regeneration of selected explants allows us to evaluate if the transformation has actually occurred; i.e., whether we will eventually obtain transformed plants. Concerning the high heterozygosis of cultivated strawberry, the best-suited explants to be targeted are adult plant materials. Here, we first used strawberry leaves from the asepsis shoots of cv. ‘Shanghai Angel’ from in vitro micropropagation of stolon apical tips. The hormonal balance is the dominant factor controlling morphogenesis in cultured explants. Thidiazuron (TDZ) has been reported to be a very efficient hormone for shoot regeneration from cultivated strawberry leaf tissues [46]. The current work revealed that the highest regeneration rate (54.8%) of asepsis materials from ‘Shanghai Angel’ was achieved by adding 2.5 mg L^−1^ TDZ to the regeneration medium (Figure 1a).

Our previous work showed that the surface-sterilized leaf explants derived from strawberry plants in a greenhouse possessed a satisfactory regeneration capability and transformability [47]. Thus, we compared the frequency and timing of shoot regeneration between explants from in vitro proliferating shoots and surface-sterilized young leaves derived from greenhouse plants of ‘Shanghai Angel’ (Figure 1b,c). On the same medium with 2.5 mg L^−1^ TDZ/0.1 mg L^−1^ indole butyric acid (IBA), a significant difference was observed between the two types of explants. After six weeks of culture, a regeneration rate of 15–20% was obtained with leaf explants from in vitro culture shoots while a drastically higher rate ranging from 50% to 100% was achieved with explants from young leaves from greenhouse plants. Clearly, the young leaves from greenhouse plants of ‘Shanghai Angel’ displayed an exceptionally high capacity for rapid shoot regeneration.

Indeed, adventitious buds from direct organogenesis were first observed on leaf discs from greenhouse plants after 3 weeks of culture on MSB5 media with TDZ/IBA at 2.5/0.1 mg L^−1^, respectively, and most explants developed adventitious buds in the subsequent 2 weeks (Figure 1c, lower panel). By contrast, only a few buds were observed on in vitro shoot-derived leaf explants after 5 weeks of culture (Figure 1c, upper panel). In addition, both direct leaf organogenesis and caulogenesis were observed on greenhouse-leaf-derived explants. However, on in vitro culture shoot-leaf-derived explants, the adventitious shoots were largely regenerated after caulogenesis. To achieve satisfactory shoot regeneration for cv. ‘Shanghai Angel’, approximately 6–8 weeks and 10–12 weeks were required for greenhouse-derived and in vitro proliferating explants, respectively.

In sum, a valid adventitious shoot regeneration system was set up from ‘Shanghai Angel’ leaf tissues with a frequency of up to 100% when explants were cultured on MSB5 medium enriched with 2.5 mg L^−1^ TDZ and 0.1 mg L^−1^ IBA. The surface-sterilized young leaf explants from greenhouse plants were found to be the most appropriate for the rapid and efficient shoot regeneration of cv. ‘Shanghai Angel’. The establishment of an efficient regeneration protocol enabled us to obtain genetically transformed strawberry without antibiotic selection.

### 2.2. Generation of a Marker-Free Binary Vector with an Intragenic Expression Cassette

The fruit-specific activity of the upstream regulatory sequence (2471 bp) of the receptacle-specific expressed KNOX family transcription factor gene03606 (FvH4_4g26090) has been demonstrated via β-glucuronidase (GUS) staining in *F. vesca* [48]. Gene 05814 (FvH4_4g15150) encodes a sugar transporter protein (STP) [49] belonging to the group of monosaccharide transporters (MSTs), which function as proton/sugar symporters for a wide range of monosaccharides [50]. The subcellular locations of FvSTP8 were predicted at the plasma membrane and the vacuolar membrane, potentially being involved in the influx of extracellular fructose and glucose in strawberry. In addition, in cultivated strawberry, this gene was preferentially expressed in functional leaves and weakly expressed in fruits [49], which is suitable for enhanced expression via genetic modification.

The generation of a marker-free binary vector for fruit-specific expression of *FvSTP8* was mainly achieved through two steps (Figure 2). We started with pMDC162 harboring the promoter of strawberry KNOX gene03606 upstream of the GUS gene [48]. First, the 1590 bp coding sequence (CDS) of *FvSTP8* was amplified from ‘Hawaii4’ cDNAs and used to replace the GUS gene downstream of the KNOX gene03606 promoter. Simultaneously, the backbone of pMDC162 beyond the GUS gene (backbone 1) was generated via reverse PCR. The recombination events of *FvSTP8* CDS and backbone 1 were screened out by PCR (*FvSTP8*-specific primers) and SpeI digestion analysis followed by sequencing confirmation with multiple T-DNA-specific primers.

A second recombination reaction was performed to substitute the natural terminator of the *FvSTP8* gene for the antibiotic marker expression cassette as well as the NOS terminator downstream of the *FvSTP8* CDS. A 401 bp fragment downstream of the stop code of *FvSTP8* was cloned from the genomic DNA of ‘Hawaii4’. At the same time, backbone 2 harboring only the gene03606 promoter and the *FvSTP8* CDS in the T-DNA region was amplified and recombined with the aforementioned *FvSTP8* natural terminator. The resultant clones were first screened out by PCR with *FvSTP8*-terminator-specific primers and then validated by sequencing with primers specific to three components of the novel intragene *FvSTP8* driven by the KNOX promoter (FvKNOX-pro: *FvSTP8*) on T-DNA. 

### 2.3. Development of Putative Intragenic Strawberry Plants via Agrobacterium-Mediated Transformation without Selection

Transformation of ‘Shanghai Angel’ began with the pre-cultured leaf explants (Figure 3a). Occasional shoot buds on explants were carefully discarded before infection with agrobacteria. After 15–20 min of infection in a freshly activated *Agrobacteria* solution, surface-dried strawberry explants were co-cultured at 22 °C for no longer than 45 h in the dark, which was crucial for suppressing the overgrowth of *Agrobacteria* on strawberry materials. After co-culture, explants were carefully washed and rinsed with antibiotics (Carbenicillin and Timentin) for the removal of *Agrobacteria*. Subsequently, explants were maintained on MSB5 medium with 200 mg L^−1^ carbenicillin and sub-cultured every 2–4 weeks (Figure 3b). Those explants with visibly resurgent *Agrobacteria* were directly discarded during the subculture, since the overgrowth of *Agrobacteria* was uncontrollable at later stages.

In the current work, once the shoot buds were apparently regenerated, plant materials were immediately transferred to a new MS medium without the plant growth regulators TDZ and IBA. Depending on the status of differentiated buds, zeatin was added (buds in health) or not added (buds in vitrification) to overcome the inhibition of TDZ on shoot elongation [51]. The regenerated adventitious shoot buds were then transferred to MS media with 6-benzyladenine (BA) and IBA at low doses to facilitate shoot growth (Figure 3c). Putatively transformed shoots through molecular validation were rooted and hardened before being transferred to pots (Figure 3d). The steps in the pipeline for *Agrobacteria*-mediated transformation of ‘Shanghai Angel’ leaf discs for marker-free strawberry are shown in Figure 3e. This method permits the development of intragenic strawberry without antibiotic selection in 3 to 6 months.

### 2.4. Selection of Transformants Using Two Rounds of PCR

In the absence of antibiotic selection, the total number of adventitious shoots regenerated from each surviving explant was very high, which is consistent with previous observations in other plant species [52]. Since the novel intragene we constructed would not express and display any visible phenotype in young shoots, it was indeed difficult to identify transformants. Here, we developed a two-step method for screening transformed plantlets. The initial step was to use a direct PCR selection of potential transformants at an early stage. This could be performed once the regenerated shoots were taller than 2 cm with three to four leaves in an in vitro culture container. At most, four shoots oriented to distinct directions of one callus were reserved and sampled for direct PCR identification.

Each regenerated adventitious shoot was sampled on a clean bench, and a coarse DNA lysis solution was prepared at 95 °C for 10 min (Figure 4a–c). This coarse DNA solution was used as a PCR template to amplify a 528 bp fragment that spanned the 3′ end of the *KNOX* promoter and the middle of the *FvSTP8* CDS with primer pairs CM2140 and CM2340 (Appendix A). PCR products were visualized on 1% agarose gel. Only those samples with clear amplicon bands identical to those of the positive control (plasmid template) were accepted as PCR-positive (Figure 4d). Unspecific amplification or *Agrobacterium* contamination might affect direct PCR with coarse DNA lysates. Actually, weak amplification was randomly observed in the direct PCR for some shoots that were later confirmed to be untransformed. In addition, the integration of backbone sequences beyond the T-DNA region is a common occurrence [29]. It is necessary to find out whether the transformants are free of backbone sequences. So, genomic DNAs were purified from candidate transformants that generated apparently strong amplicons in the initial direct PCR and were used as templates for a second PCR analysis targeting both T-DNA and the vector backbone. As shown in Figure 4e, the two independent lines tested were true transformants, but were not free of the vector backbone.

To summarize, in this work ‘Shanghai Angel’ leaf explants were infected with *Agrobacterium* in four independent experiments using greenhouse-derived young leaves and those from in vitro culture after 2–4 weeks of pre-culture (Table 1). Through *Agrobacterium* co-culture of less than 2 days followed by subculture without selection, a total of 241 shoots regenerated in five to six months after the initial infection. Indeed, more than half of the explants were discarded during subculture due to the resurgent growth of *Agrobacterium*. The surviving materials often developed bushy shoots on the explant. Then, direct PCR identification was performed for all shoots from the four transformation trials. A total of four independent shoots were identified as PCR-positive. Therefore, expressing the transformation efficiency as the percentage of PCR-positive shoots over the number of initial leaf disc explants, the maximum transformation efficiency achieved in this work was 5.26%. If the transformation frequency is expressed as the percentage of PCR-positive shoots over the number of tested shoots, the maximum transformation efficiency was about 0–2.38%.

### 2.5. Determination of the Intragene Copy Number in Transformants Using ddPCR

Other than direct PCR to accurately identify transformants, probe-based duplex ddPCR was further used to estimate the intragene integrations in the four independent T0-transformed events of ‘Shanghai Angel’. Primers and probes were designed to detect the reference gene *FaDHAR* (DeHydro Ascorbate Reductase) and the intragene *FvSTP8* driven by the KNOX promoter (Figure 2). *FaDHAR* is a low-copy nuclear gene [53] with four copies per octoploid strawberry genome [4]. Using the updated DNA sequence information in *Fragaria × ananassa* Reference Genome v1.0 (FANhybrid_r1.2) at the GDR (www.rosaceae.org, accessed on 20 June 2021) [54], reference primers and probes (Appendix A) were designed to detect the single-copy FxaC_26g17540 (FANhyb_rscf00000137.1.g00005.1) in the octoploid strawberry genome.

DdPCR was independently performed twice with untransformed ‘Shanghai Angel’ as a negative control (Figure 5). Among the different ddPCR assays, 20,072–21,444 droplets (mean: 20,739) were generated per reaction. Nearly 7386–10,772 (mean: 9291) positive droplets for the reference amplicon as well as a varying number of positive droplets for the intragene amplicon were quantified per 20 μL reaction harboring 5 ng of fractioned DNA template. The original one-dimensional plots of droplets for all four transformed lines and the blank sample (untransformed ‘Shanghai Angel’) indicated that the droplet patterns were largely regular for an accurate quantification in this study (Appendix A). The typical two-dimensional (2-D) plot of droplets measured in line #13-1 for fluorescence signals emitted from both the reference gene (blue, FAM-labeled) and the target intragene (red, VIC-labeled) regions clearly showed the general pattern of ddPCR (Figure 5a). The 2-D droplet patterns for other lines and the blank control are displayed in Appendix A. The number of VIC-labeled positive droplets of the intragene in transformants dramatically ranged from 55 to 19,471 (Figure 5b), while the blank sample of untransformed ‘Shanghai Angel’ within each run displayed a background VIC amplitude including four positive droplets across this study (data omitted).

The ddPCR results for four lines selected from the direct PCR are shown in Figure 5c. The intragene copy numbers in #9-d and #20-g were significantly lower than 1, suggesting the chimeric status of these two lines. Chimerism occurred at a very high frequency (about 50%) in this study, and the two chimeras were not used in further experiments. By contrast, the DNA insertion values in #13-1 and #20-7 were particularly high: close to 6 and 11 copies, respectively. The latter two lines were recovered and maintained. It is a pity that plantlets harboring a low copy number (average: 1–2 copies) of the novel intragene were not obtained and await to be identified in the near future.

## 3. Discussion

The cultivar ‘Shanghai Angel’ is characterized by a superior resistance to *Colletotrichum* spp., the causal agents of anthracnose, and a high yield with satisfactory fruit quality under most conditions [55,56]. However, in a climate full of rainy and sunless days, which frequently occur during early spring in Shanghai, the fruit quality of this variety declines sharply. To increase the fruit quality of cv. ‘Shanghai Angel’ in an unfavorable climate, one possible approach is to introduce the fruit-specific expression of a sugar transporter gene mediating the influx of extracellular sugar into fruits. The current work aimed to develop consumer-friendly intragenic octoploid strawberry. Accordingly, we constructed a marker-free vector using only DNA fragments of woodland strawberry origin to produce strawberry with fruit-preferential expression of a sugar transporter. Based on an efficient regeneration protocol, *Agrobacterium*-mediated transformation of cv. ‘Shanghai Angel’ without antibiotic selection, we successfully introduced a novel intragenic expression cassette FvKNOX-Pro: *FvSTP8* into this cultivar. However, the few complete transformants obtained have integrations of the vector backbone beyond the T-DNA region, indicating that they are not true intragenic strawberry. Clearly, the way to true intragenic strawberry is long and winding.

Despite the current frustration, we hope to obtain the desired intragenic strawberry with a single T-DNA insertion and without the backbone in the near future, when more transformants could be developed. Still, the described transformation system and intragenic vector construction approach could be applied to create novel strawberry germplasms and are potentially useful in bringing wild-type genetic diversity back to cultivated strawberry. In addition, a ddPCR assay was developed for octoploid strawberry in the present study, thereby enabling accurate monitoring of DNA integration into the engineered strawberry or the detection of mutations in a germplasm population at an affordable cost.

Results of Southern blot analyses are sometimes complex and do not allow for an accurate determination of the genomic integration pattern, especially for multiple insertions at a single locus and when the hybridized signal is influenced by other factors [44,57]. The genome size of cultivated strawberry is 813 Mb [1], and one copy genome is ca. 0.89 pg (https://cvalues.science.kew.org/, accessed on 20 June 2021). According to a previous suggestion on the appropriate amount of template using the BioRad ddPCR system [44], we started with 25 ng of strawberry genomic DNA template in ddPCR using Sniper DQ24. However, there occurred too many positive droplets to allow for accurate counting. When we reduced the template amount to 4~5 ng, a satisfactory determination of positive and negative droplets was achieved. This inconsistency might be caused by different levels of effectiveness in sample amplification and/or florescent signal acquisition in the two different systems. 

Direct PCR combined with ddPCR revealed a maximum transformation efficiency of 5.3% for cv. ‘Shanghai Angel’ without selection in this study, although the four independent transformants obtained were composed of two chimeras and two transformants with multiple-copy integration. The transformation frequency of strawberry varies greatly and is affected by the cultivar and methods. The efficiency reported in early studies (before 2005) was for the most part less than 10%. The cultivars ‘Rapella’ [58] and ‘Redcoat’ [59] were transformed at a frequency of 0.95% and 3%, respectively. A transformation of the strawberry cv. ‘Firework’ using *Agrobacterium* strain CBE21 for an overexpression of the traumatin II gene achieved an efficiency of about 11% [60]. Later, a work using LBA4404 *Agrobacterium*-mediated transformation of cv. ‘Chandler’ reached an extraordinarily high efficiency ranging from 40% to 90%, although a high frequency of non-TNA backbone integration in the transformed strawberry was also observed [61]. It was reported that the octoploid strawberry genotype ‘Laboratory Festival #9′ (LF9) displayed close to 100% transformability under 5 mg/L Kan selection [46].

Here, a striking difference was found in the regeneration frequency of leaf disks from greenhouse-derived young leaves with those from in vitro sterile shoots of the cv. ‘Shanghai Angel’. Apparently, greenhouse-derived young leaf materials exhibit a faster and greater capability to regenerate than in vitro materials. Consistently, transformation with leaf pieces from ‘Chandler’ seedlings was successful at a frequency of 4.16% with differentiated plantlets, while undifferentiated and transformed callus was obtained at a frequency of only 1.11% with leaf disks from micropropagated shoots [62]. Additionally, an increased transformation efficiency might be obtained by optimizing the use of antibiotics for the satisfactory suppression of the resurgence of agrobacteria, since in the current case more than half of the explants with *Agrobacteria* contamination were discarded during subculture.

Reducing the rate of chimerism in transformed shoots is a similar bottleneck in genetic engineering for strawberry and other *Rosaceous* plants [63]. To dissociate chimeras and recover completely transformed shoots in a transformation without selection is a great challenge. A high rate of chimerism occurred in this study, since when there was no selection or the selection applied was low, non-transformed cells would escape and the chimeras would survive. In addition, the chimerism rate is associated with the regeneration pathways of explants, and there is a high risk of obtaining chimeric plants from organogenesis [64,65]. Clearly, the leaf explants of the cv. ‘Shanghai Angel’ directly harvested from the greenhouse showed a higher organogenic capacity than those from in vitro culture shoots. The independent transformed lines #9-d and #20-g obtained in this study were chimeras, and the possibility could not be excluded that these seedlings might be regenerated from transformed cells via organogenesis. Actually, as shown in Figure 1c, most leaf explants of cv. ‘Shanghai Angel’ obtained from micropropagation regenerated via caulogenesis on wounding sites, which would reduce the frequency of chimerism during transformation. Therefore, we suggest the use of strawberry plants micropropagated in vitro as donor plants for genetic transformation.

The transformation of a plant is actually a war of cell survival, where the plant’s defense system interacts with the intruding pathogen *Agrobacterium* [63]. To obtain a desirable transgenic or intragenic plant with a single copy of a novel DNA is a matter of the two organisms. Although the disadvantage of multiple inserts during plant transformation is well known, multiple inserts often occur in *Agrobacterium*-mediated transformation. In the three independent lines of marker-free transgenic apple, the copy number of the target gene ranges from 2 to 5 [21]. In a transformation of the wheat cv. ‘Fielder’, only 24% of the lines contained a single copy of the transgene, while the remaining lines harbored two or more copies of T-DNA insertions, with a maximum of seven copies [19]. Concerning the frequency of T-DNA insertions in *Agrobacterium*-transformed strawberry, various observations have been reported. According to a Southern blot assay, a high frequency (four of six) of single-copy T-DNA integration was found in the transgenic strawberry cv. ‘Calypso’ [20]. However, while the Southern blot provided evidence for transformation or the potential for low copy numbers of T-DNA inserts, it did not determine the copy number of DNA insertions in strawberry of the other two cases [10,66]. In previously engineered strawberries, the copy numbers of T-DNA inserts ranged from 1 to 5, markedly varying with the strawberry variety and the *Agrobacterium* strain used [61,67,68,69,70,71]. Apparently, *Agrobacterium* strains have the distinct capacity to deliver T-DNA into different hosts, even plants of the same species. In developing marker-free banana, LBA4404-mediated transformation resulted in an average of two copies of the transgene in two cultivars, while AGL1 generated very high copy numbers: an average of 5 and 13 in these varieties, respectively [22]. Whether the *Agrobacterium* strain LBA4404 or EHA105 is less virulent than GV3101 to the strawberry cv. ‘Shanghai Angel’ needs to be revealed in the future. 

In addition to studying the effect of enhanced fruit expression of *FvSTP8* in octoploid strawberry via qualitative evaluation of the transformed plants, our future work will focus on testing different *Agrobacterium* strains and adjusting the use of antibiotics and hormones in order to increase the competence of both strawberry and *Agrobacterium* for transformation and orchestrate their survival. The final goal is to improve the transformation efficiency and develop true intragenic plantlets free of the vector backbone while reducing the frequency of chimerism and multiple inserts in strawberry transformants.

## 4. Materials and Methods

### 4.1. Plant Materials and Growth Conditions

The strawberry cv. ‘Shanghai Angel’, released by the Shanghai Academy of Agricultural Sciences, is a progeny of the cross ‘Benihoppe’ × ‘Akihime’ and represents an elite cultivar for Shanghai and eastern China. For strawberry transformation, leaves from cv. ‘Shanghai Angel’ plants in both a greenhouse and tissue culture were utilized. Strawberry cv. ‘Shanghai Angel’ plants were grown in a greenhouse under a 10 h photoperiod with a light intensity of 200 μM m^−2^ s^−1^ at 25 °C (day)/22 °C (night) in a mixed substrate (peat: coconut husk: pearlite: vermiculite in a volume ratio 2:1:1:1). For tissue culture, shoots were developed from the apical tips of stolons. Stolon tips were cut into pieces of around 10 cm in length from greenhouse-derived plants, thoroughly cleaned with a brush and a droplet of dishwashing liquid, and then washed under running tap water for 1 h. Later, stolon tips of around 5 cm in length were surface-sterilized via soaking them twice in 4% (*v*/*v*) commercial sodium hypochlorite solution (Cat. No. S1944-500 mL, 5.68% active chlorine, Sangon, Shanghai) and gently agitating them for 5 min each time, followed by eight rinses with sterile water. Apical tips of around 0.5 cm were cut from the sterilized stolon tips on sterile filter paper and placed on MS medium with 0.1 mg L^−1^ Gibberellic acid (GA, Cat. No. A600738, Sangon, Shanghai) and 0.3 mg L^−1^ BA until germination. The micropropagated shoots were subcultured every 6 to 8 weeks on MS medium with BA 0.3 mg L^−1^ and IBA 0.01 mg L^−1^. The subculture times for asepsis shoots were limited to less than 5 to ensure a sufficient capability for regeneration in the leaf discs.

To identify the optimal hormonal conditions for regeneration of ‘Shanghai Angel’, leaf discs from in vitro asepsis shoots were cut and grown on a range of MSB5 media enriched with 0.1 mg L^−1^ IBA as well as TDZ that varied in concentration from 2.0 to 3.0 mg L^−1^. Incubation in the dark for 2 weeks followed by a photoperiod with a weak light intensity of 40–60 μM m^−2^ s^−1^ at 25 °C (12 h day/12 h night) was uniformly used for all treatments. Plant explants were subcultured on the same media every 30 d. The effect of TDZ supplemented in MSB5 media on shoot regeneration was evaluated after 60 d of culture.

### 4.2. Agrobacterium Strain and Binary Vector

The binary plasmid pMDC162 harbors the promoter of strawberry KNOX gene03606 enabling receptacle-specific GUS activity [48]. The original pMDC162 vector (pMDC162-P03606:GUS) was modified mainly through two recombination reactions. First, the *FvSTP8* CDS was cloned from leaf cDNAs of *F. vesca* ‘Hawaii4’ into pClone007B blunt vector (TSV-007B, Tsingke, Beijing) through RT-PCR and validated by sequencing. The *FvSTP8* CDS was reamplified with primers CM2268 and CM2269 in order to replace the GUS gene downstream of the KNOX gene03606 promoter. The 11.77 kb backbone of pMDC162 without the GUS gene (for short, backbone 1) was generated via reverse PCR using Phanta EVO Super-Fidelity DNA Polymerase (P503-d1, Vazyme, Nanjing, China) with primers ZT067 and ZT068. Then, a *FvSTP8* CDS fragment and the pMDC162-P03606 backbone 1 were recombined using the Exnase II from the ClonExpress II One Step Cloning Kit (C112, Vazyme, Nanjing, China). The resultant clones (for short, pMDC162-P03606:FvSTP8) were first identified via PCR with FvSTP8-specific primers combined with restriction enzyme SpeI digest analysis, since the original pMDC162 plasmid harbors two recognition sites but the resultant novel construct has three sites. Furthermore, one of the resultant plasmids was selected via sequencing with eight specific primers spanning the whole T-DNA region.

To obtain a marker-free plasmid, a second recombination was designed to discard the antibiotic marker gene expression cassette in pMDC162-P03606:*FvSTP8*. A 401 bp natural terminator of the *FvSTP8* gene (directly downstream of the stop code of *FvSTP8*) was cloned from the genomic DNA of ‘Hawaii4’ into pClone007B vector. After sequencing for validation, this terminator was reamplified with primers CM2328 and CM2329. Simultaneously, the backbone of pMDC162-P03606:FvSTP8 without the NOS terminator behind the *FvSTP8* CDS as well as the whole HygR expression cassette (CaMV35S: HygR gene-CaMV polyA signal) (for short, backbone 2) was amplified via reverse PCR using primers CM2269 and ZT069. The FvSTP8 terminator and the above backbone 2 were recombined using the Exnase II of the C112 kit (Vazyme, Nanjing, China). Clones were first screened by PCR with FvSTP8 terminator-specific primers and then validated by sequencing with three specific primers. Finally, the resultant plasmid was named pMF-P03606:*FvSTP8*, which was introduced into *Agrobacterium* strain GV3101 by the freeze–thaw method [72]. Primers used to generate pMF-P03606:*FvSTP8* are listed in Appendix A.

### 4.3. Explant Preparation and Strawberry Transformation

Newly unfolded trifoliate leaves of strawberry cv. ‘Shanghai Angel’ were obtained from the greenhouse and similarly disinfected as stolon tips. Leaves were cut into discs of around 1 × 1 cm (greenhouse leaves) or 0.3 × 0.5 cm (in vitro culture leaves) and oriented with the adaxial side of the explant on the regeneration medium. Pre-culture and shoot regeneration from leaf disks were achieved on MSB5 (MS basal salts with B5 vitamins, M404, PhytoTech, USA) (pH 5.7–5.8) containing 20 g L^−1^ sucrose, 7.5 g L^−1^ plant agar (A800728, Macklin, Shanghai, China), and 2.5 mg L^−1^ thidiazuron (TDZ) (A600746-0025, Sangon, Shanghai) together with 0.1 mg L^−1^ 3-indole butyric acid (IBA) (A600725-0025, Sangon, Shanghai) [73]. Cultivation for two weeks in darkness and then three to four weeks with a low light intensity (about 40–60 μM m^−2^ s^−1^) at 25 ± 1 °C was carried out during pre-culture.

*Agrobacterium* strain GV3101 harboring the target marker-free construct was cultured overnight in 4 mL of Luria Broth (LB) liquid containing Kanamycin 50 mg L^−1^, rifampicin 20 mg L^−1^, and gentamycin sulfate 25 mg L^−1^ at 28 °C in a shaker at 250 rpm. Transformation was largely performed as previously reported [74]. The overnight culture was centrifuged at 5000 rpm for 8 min and the pellet was suspended in liquid MSB5 medium (M404, PhytoTech, USA) pH 5.5, 2% sucrose, and 100 μM acetosyringone (AS, Cat. No. A601111, Sangon, Shanghai) (OD_600nm_ ≈ 0.1) in a flask and gently activated for 2–3 h at 25 °C in darkness in a shaker at 50 rpm. The pre-cultured explants were immersed into the *Agrobacterium* suspension for 15–20 min on a clean bench. After infection, the explants were dried on sterile filter paper and cultivated on an MSB5 medium plate with 2% sucrose, 100 μM AS, 2.5 mg L^−1^ TDZ, and 0.1 mg L^−1^ IBA at a pH of 5.5 in darkness at 22 °C for about 45 h. After co-cultivation, the explants were soaked in sterile water containing 200 mg L^−1^ Carbenicillin and 200 mg L^−1^ Timentin for 15–30 min, rinsed three times, and then dried on filter paper. After washing, the explants were transferred to MSB5 as a solid shoot induction medium with 2% sucrose, 200 mg L^−1^ Carbenicillin, 2.5 mg L^−1^ TDZ, and 0.1 mg L^−1^ IBA at a pH of 5.8. Subculture was carried out every 2–4 weeks on the same medium until shoots were generated. The bushy shoot buds were transferred to MS medium (M519, PhytoTech, USA) without any hormones (if vitrification happened) or with 0.22 mg L^−1^ Zeatin (no vitrification) to facilitate shoot development [51]. After cultivation on MS without plant hormones or with ZT only once, the bushy shoots were then transferred to MS medium with BA 0.1 mg L^−1^ and IBA 0.01 mg L^−1^ to enhance shoot elongation.

### 4.4. Direct PCR Analysis of Regenerated Shoots

Before rooting and acclimation, independent transformants were first genotyped using direct PCR. A piece of round leaf disc with a diameter of 3 mm was sampled from each candidate transformant on a clean bench and immersed into 50 μL of buffer A (100 mM NaOH, 2% Tween-20) where it was lysed at 95 °C for 10 min. This coarse lysate was then mixed with 150 μL of buffer B (Tris-HCl 100 mM, EDTA-2Na 2 mM, pH 2.0) for dilution and the resultant solution was directly used as a template for PCR.

Specific primers matching to a junction region of the T-DNA in the pMF-P03606:*FvSTP8* construct were designed to screen the transformed plantlets. Amplification with strawberry *Actin* (gene 18570-v1.0 hybrid, *FvH4_1g23490*) specific primers 5′-CGACCTTAATCTTCATGCTGCTTGGA-3′ and 5′-TCATTGGAATGGAAGCTGCTGGCATT-3′ [75] was performed to ensure the quality of templates used for direct PCR. The PCR reaction cocktail contained 1 μL of the above coarse sample lysate, 0.3 μL of each primer (stock: 20 μM), 0.5 μL of dNTP (stock: 2.5 mM), 2 μL of 10× PCR buffer with Mg2+, and 0.3 μL of Taq DNA polymerase (5 U/μL, Biocolor, Shanghai, China). A second PCR was further performed to validate the potential transformants using primers matching both the T-DNA and non-T-DNA vector backbone regions. The primers used for direct PCR screening and the second validation PCR are displayed in Appendix A. Reactions were carried out in a program consisting of an initial denaturation at 94 °C for 3 min, followed by 32 cycles of 94 °C for 30 sec, 58 °C for 30 sec, 72 °C for 40 sec, and a final extension at 72 °C for 5 min. PCR products were separated using 1.0% agarose gel electrophoresis, visualized with GelRed dye under UV light, and photographed with a Tanon 1600 Gel Photographer (Tanon, Shanghai, China).

### 4.5. DNA Purification and ddPCR Analysis

Total strawberry genomic DNA was isolated from leaves of transformed plantlets and from untransformed plants using a Super Plant Genomic DNA kit for polysaccharides and polyphenolics-rich samples following the manufacturer’s instructions with minor modifications (TianGen, Beijing, Cat. No. DP360). Briefly, strawberry leaves were homogenized at 55 Hz for 1 min using Tissue Lyser-24 (Jingxin, Shanghai). Twenty to forty milligrams of plant powder was mixed with 400 μL of GPA buffer followed by water bathing at 65 °C for 30 min. The purified DNA was visualized using 1% agarose gel electrophoresis and then quantified using a Qubit 4.0 Fluorometer (Thermo Fisher Scientific, Waltham, MA, USA) and the dsDNA quantification kit (Magic dsDNA HS Assay Kit, Cat. No. Q32854, Magic Biotech, Hangzhou, China). One hundred nanograms of genomic DNA per sample was fractionated with 8 U of SpeI-HF (NEB, R3133 L) at 37 °C for 30 min. Four or five nanograms of DNA after digestion was used for each dPCR reaction. A negative control (blank) was performed using untransformed ‘Shanghai Angel’ DNA for each new ddPCR assay.

The low-copy nuclear gene DeHydro Ascorbate Reductase (DHAR) (four copies per octoploid genome) previously used for phylogenetic analysis in the genus *Fragaria* [53] was chosen as the reference gene for the current ddPCR. For the probe-based duplex ddPCR, primers and probes were designed using Primer3Plus: http://www.bioinformatics.nl/cgi-bin/primer3plus/primer3plus.cgi/, accessed on 20 June 2021 [76]. Candidate primer pairs and probes were further selected through evaluation at https://mfeprimer3-0.igenetech.com/, accessed on 20 June 2021 [77] following the criteria suggested previously [44]. The sequence information on primers and probes for ddPCR is given in Appendix A. Reference-gene-specific primers and probes were designed to match only FxaC_26g17540, a copy of DHAR corresponding to the loci FANhyb_rscf00000137.1.g00005.1 in *Fragaria × ananassa* Reference Genome v1.0 (FANhybrid_r1.2) scaffolds at the GDR (www.rosaceae.org, accessed on 20 June 2021).

A ddPCR mixture was prepared with 10 μL of 2×TIANexact Genotyping qPCR Pre Mix (probe) (no dUTP; Cat. FP211, Beijing, China), 450 nM of each primer pair, 250 nM of each probe for both the reference gene and the intragenic construct, and the 5 ng of digested genomic DNA in a final volume of 20 μL. Droplet generation (Vibro Ject injection technology), PCR amplification, fluorescence detection, droplet reading, and quantification were accomplished using a DQ24 digital PCR machine (Sniper, Suzhou, China). The program was set at an initial 65 °C for 5 min, 95 °C for 5 min, and 40 cycles of 95 °C for 20 s and 63 °C for 30 sec. Following the counting of droplets, measurements of the intragene copy number were performed using the Sniper SightPro software with default settings for the threshold to separate positive and negative droplets. DdPCR was independently repeated twice and similar results were obtained. Results presented are from one of the experiments in which the positive and negative signal intensities were separated most satisfactorily (Appendix A). The average frequency (λ) of certain molecules in ddPCR was calculated following the equation: λ= −ln (1 − positive droplets/total droplets)/volume of droplet. The copy number (C) of the intragene per genome was obtained using the equation C = 1 * λ_target/_λ_ref_, where the constant 1 represents the copy number of the detected reference *FaDHAR* region per genome, λ_target_ represents the average frequency of the intragene KNOX-Pro:*FvSTP8*, and λ_ref_ represents the average frequency of the detected *FaDHAR* region.

## Figures and Tables

**Figure 1 plants-10-02229-f001:**
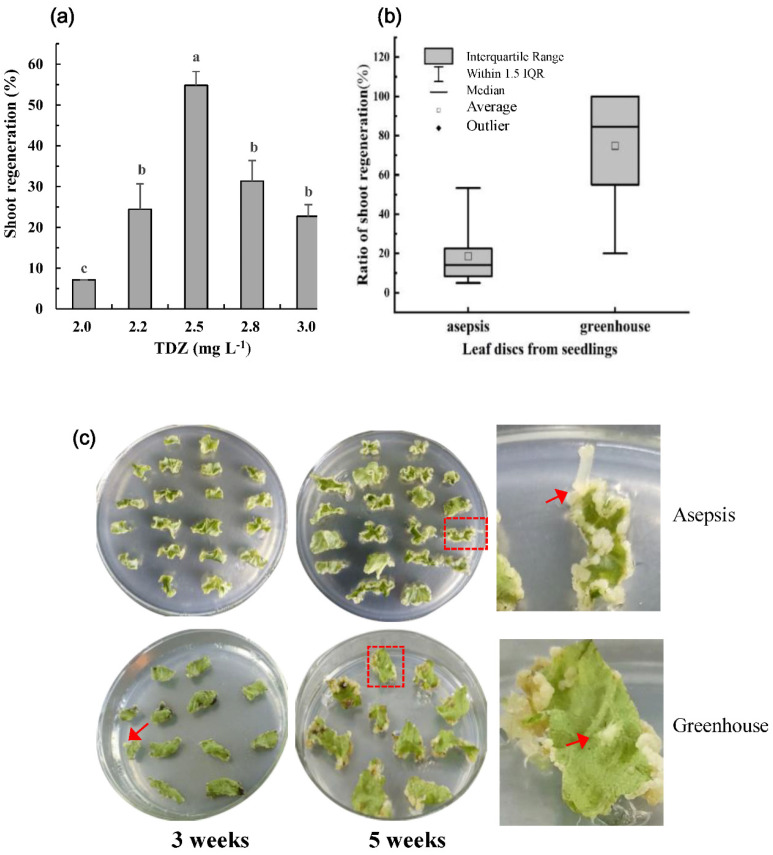
Efficient shoot regeneration of the strawberry cv. ‘Shanghai Angel’. (**a**) The frequency of regeneration from asepsis leaf discs after 60 days’ cultivation on media with thidiazuron (TDZ) at different concentrations. The same MSB5 medium with 0.1 mg L^−1^ IBA was used for all treatments. Different letters marked over columns indicate significant differences among treatments (Duncan’s test, *p* ≤ 0.05). (**b**) The box chart indicates the comparative shoot regeneration of leaf discs from asepsis shoots and greenhouse plants. Data were obtained after six weeks’ cultivation on MSB5 with 2.5 mg L^−1^ TDZ and 0.1 mg L^−1^ IBA. (**c**) The typical morphology of leaf discs after three weeks and five weeks of cultivation on the same media. Upper, asepsis-shoot-derived leaf discs; lower, greenhouse-plant-derived leaf discs; right, enlarged leaf discs squared. Arrows indicate the shoot buds differentiated either from callus or direct adventitious organogenesis.

**Figure 2 plants-10-02229-f002:**
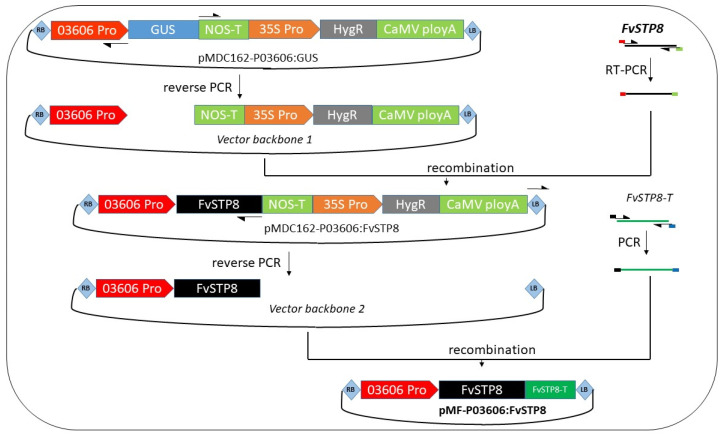
The T-DNA regions of the 12.5 kb binary vector pMDC162-P03606:*GUS* and its derivative pMF-P03606:*FvSTP8* (11.1 kb). Two recombination reactions were sequentially performed. First, the *GUS* gene was substituted with the *FvSTP8* coding sequence (CDS) from ‘Hawaii4’. Then, the NOS terminator behind the *FvSTP8* CDS was replaced with the original terminator of *FvSTP8* from ‘Hawaii4’ and the whole expression cassette for the hygromycin selection marker was simultaneously deleted. LB/RB, left/right border; STP8, coding sequence of *FvSTP8* from “Hawaii4”; STP8-T, terminator of *FvSTP8* from “Hawaii4”; NOS-T, terminator of the *A. tumefaciens NOS* gene; 35S, CaMV *35S* promoter; HygR, hygromycin resistance gene with a CaMV *35S* promoter and a CaMV polyA signal terminator. Arrows indicate the primers for PCR or reverse PCR; colors indicate the primers for the overlapping sequences used in recombination.

**Figure 3 plants-10-02229-f003:**
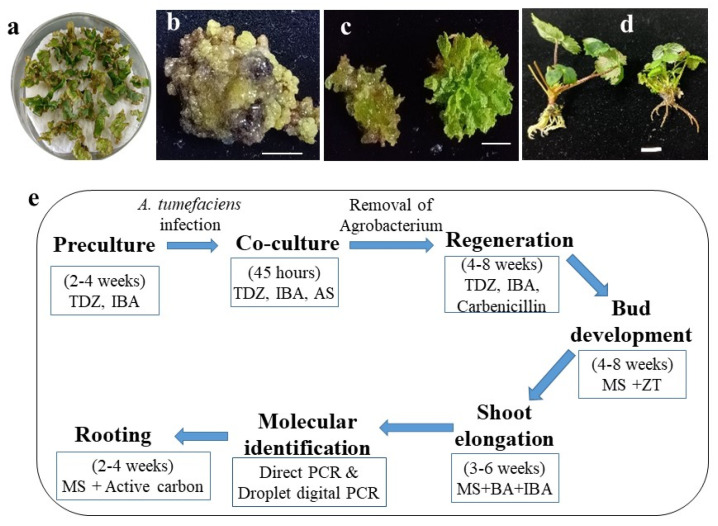
*Agrobacterium*-mediated transformation of the strawberry cv. ‘Shanghai Angel’ without selection. (**a**) Explants were rinsed with sterile water containing 200 mg L^−1^ Carbenicillin and 200 mg L^−1^ Timentin after 45 h’ co-culture at 22 °C in darkness. (**b**) Transformed calli undergoing differentiation on MS medium with B5 vitamins (MSB5), 2.5 mg L^−1^ TDZ, 0.1 mg L^−1^ IBA, and 200 mg L^−1^ Carbenicillin. (**c**) The bushy buds and shoots after regeneration. (**d**) Two transformed plantlets validated by PCR were washed before being transferred into the soil substrate. Scale bars in (**b**,**c**), 0.5 cm; scale bar in (**d**), 1 cm. (**e**) Brief scheme of the methodology followed for the transformation of strawberry with the marker-free vector pMF-P03606:*FvSTP8*. MSB5 medium was used in the previous three steps; MS medium was used after shoot regeneration. AS, acetosyringone at 100 μM to facilitate *Agrobacteria* (strain GV3101) infection; ZT, zeatin at 0.22 mg L^−1^ to enhance bud development; BA 0.1 mg L^−1^ and IBA 0.01 mg L^−1^ for shoot elongation; active carbon at 0.4 mg L^−1^ for rooting.

**Figure 4 plants-10-02229-f004:**
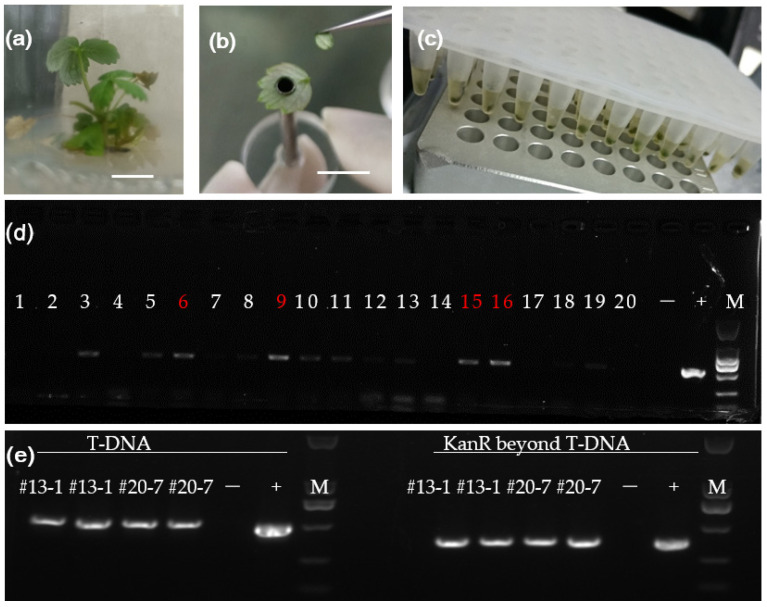
Direct PCR identification of the sterile shoots potentially transformed with pMF-P03606:*FvSTP8*. (**a**) One regenerated young plantlet. (**b**) A piece of leaf with a diameter of 3 mm was sampled for direct PCR analysis. Scale bars, 1 cm for (**a**,**b**). (**c**) Coarse lysis solution served as a DNA template for direct PCR analysis. (**d**) Gel photo showing the amplification of the intragenic construct in one direct PCR analysis of 20 plantlets. Lane 6 for #9-d; lane 9 for #13-1; and lanes 15 and 16 for #20-g and #20-7, respectively, with 1.0% agarose gel at 140 V for 15 min. (**e**) A second PCR analysis of the purified DNA from potential transformants against both a T-DNA insert and the vector backbone. The amplicons specific to T-DNA (with CM2140 and CM2340 primers) and KanR beyond T-DNA were 528 bp and 400 bp, respectively. PCR template ‘−’ for untransformed ‘Shanghai Angel’ and ‘+’ for pMF-P03606:*FvSTP8* plasmid. M, DL2000 DNA marker.

**Figure 5 plants-10-02229-f005:**
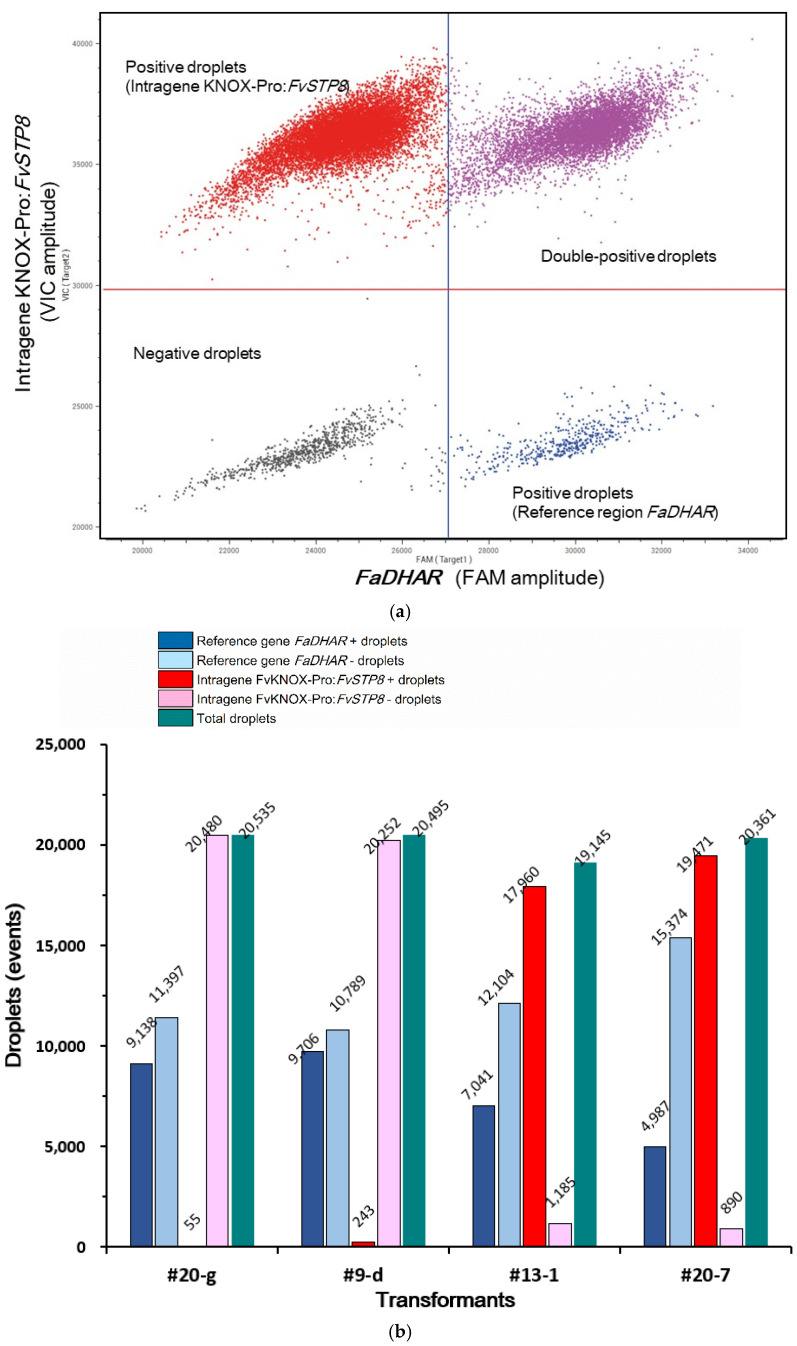
Intragene copy number measurement in the strawberry cv. ‘Shanghai Angel’ via ddPCR. (**a**) Two-dimensional plots of the fluorescence signal emitted from the endogenous reference gene *FaDHA*R (FAM^TM^ labeled, blue positive droplets) and the intragene KNOX Pro:*FvSTP8* (VIC^TM^ labeled, red positive droplets) for line #13-1. Negative droplets are black. The droplets with both fluorescent probes are purple. (**b**) Bar graph highlighting the relative abundance of positive and negative droplets relative to the total number of droplets obtained in the four independent lines. (**c**) Copy number of the intragene in T0 strawberry lines after variation processing in Sniper SightPro software, where the reference gene copy was set to 1 in the octoploid genome based on BLASTN analysis. The error bars represent the maximum and minimum Poisson distribution for the 95% confidence interval generated by the Sniper SightPro software.

**Table 1 plants-10-02229-t001:** Transformation efficiency with pMF-P03606:*FvSTP8* of the strawberry variety ‘Shanghai Angel’ without antibiotic selection and as revealed by PCR analysis.

Experiment ^a^	No. of Initial Explants	No. of Shoots ^b^	No. of PCR-Positive Shoots	Transformation Efficiency (%) ^c^
I	30	53	0	0
II	51	58	1	1.96%
III	40	46	1	2.50%
IV	38	84	2	5.26%

^a^ In experiment I, micropropagated shoot-derived leaves were used as explants, while in the rest of the experiments greenhouse-derived leaf explants were used. ^b^ The number of regenerated shoots was calculated at five to six months after transformation. ^c^ Transformation efficiency is expressed as the percentage of PCR-positive shoots over the number of initial explants.

## Data Availability

The data obtained in this study are available in this article.

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
