# Peer review of "A Strategy for the Production and Molecular Validation of Agrobacterium-Mediated Intragenic Octoploid Strawberry"

_plants, 2021, doi:10.3390/plants10112229_

Round 1
Reviewer 1 Report
The authors focused on production of marker free transgenic strawberry plants. Strawberry explants were co-cultivated with Agrobacterium tumefaciens and subsequently regenerated under non-selective conditions. Transgenic shoots were selected based on the molecular analyses (direct PCR and duplex droplet digital ddPCR). Using this approach, 4 out of 241 regenerated shoots were evaluated as transgenic. Two shoots were chimeras and 2 shoots contained more than 6 transgene copies.
The structure of the manuscript is a bit disordered. The results do not show what is stated in the title. Experiments such as TDZ concentration evaluation or the use of aseptic/greenhouse leaves (Fig. 1) are not important for the focus of the MS. The authors should have solved it before. Too much space is devoted to preparing a vector construct, Fig. 2 should be moved to suppl material. I am not quite sure whether the described approach really leads to the successful obtaining of shoots, as identified transgenic shoots are i) with incomplete T-DNA or ii) the T-DNA copy number is too high for the use in standard transgenosis. In the case of standard regeneration conditions (under selection pressure), such shoots would not regenerate at all.
Figures are not self-explanatory. The quality of Fig. 4d is very bad, the lanes are unidentifiable (positive/negative control). Moreover, it seems that the bands in the lanes 3 and 4 are quite bigger size than the band in the lane 2.
Author Response
Reviewer 1
The authors focused on production of marker free transgenic strawberry plants. Strawberry explants were co-cultivated with Agrobacterium tumefaciens and subsequently regenerated under non-selective conditions. Transgenic shoots were selected based on the molecular analyses (direct PCR and duplex droplet digital ddPCR). Using this approach, 4 out of 241 regenerated shoots were evaluated as transgenic. Two shoots were chimeras and 2 shoots contained more than 6 transgene copies.
Q1: The structure of the manuscript is a bit disordered. The results do not show what is stated in the title. Experiments such as TDZ concentration evaluation or the use of aseptic/greenhouse leaves (Fig. 1) are not important for the focus of the MS. The authors should have solved it before. Too much space is devoted to preparing a vector construct, Fig. 2 should be moved to suppl material.
A1: Thank Reviewer for the comment. The choice of explant and a robust regeneration method are important for the efforts to develop intragenic transformant without selection. It is true as Reviewer suggested, that we have solved it at the beginning. Since this fundamental research is relevant to the development of transformants and the chimerism state for some transformants, we believe it deserve to be presented in the Result section. Concerning the vector used in current work, it was different from previous reported marker-free plasmids. We think it better to clarify the process for developing a novel vector for the intragenic strawberry. We really appreciate your understanding. Thank you.
Q2: I am not quite sure whether the described approach really leads to the successful obtaining of shoots, as identified transgenic shoots are i) with incomplete T-DNA or ii) the T-DNA copy number is too high for the use in standard transgenosis. In the case of standard regeneration conditions (under selection pressure), such shoots would not regenerate at all.
A2: The transformants we obtained in current work actually are not the desirable with a single copy insertion, which is really as Reviewer commented. The regeneration pathway of explants (organogenesis), no selection pressure, and the high virulence of Agrobacterium strain used, jointly increased the frequency of chimerism and multiple insertions in our transformants. In future work we will keep on optimizing the transformation for marker-free intragenic strawberry as we discussed in the Discussion section. Thank Reviewer very much for your valuable comment.
Q3: Figures are not self-explanatory. The quality of Fig. 4d is very bad, the lanes are unidentifiable (positive/negative control). Moreover, it seems that the bands in the lanes 3 and 4 are quite bigger size than the band in the lane 2.
A3: We are very sorry for the incorrect presentation in previous submission. We have redone this analysis from the beginning. After sampling and a fast lysis, a direct PCR was performed followed with an agarose gel electrophoresis. Please see the novel Figure 4d in Page 12. Special thanks to Reviewer for this critical comment.
Reviewer 2 Report
The manuscript describes a study aimed to obtain intragenic octoploid strawberry improved for quality trait under unfavorable climate, through the optimization of in vitro culture protocol and Agrobacterium-mediated intragenic transformation system to produce marker-free transformed plants. The results obtained are interesting but could be described in a more concise and clear manner. Furthermore, the text can be improved, taking care to correct or explain some inconsistencies.
Extensive English revision by a native speaker is also required, making the manuscript easier to read.
More suggestions are given to the authors and added directly in the attached text.
Title: “strategy” maybe more suitable than “method”.
Abstract: The Abstract is too long. According to the Instructions for Authors it should be a total of about 200 words maximum.
Results: In this Section only the results must be described, also in a robust and clear way. Actually, there is a lot of information that is more suitable for Discussion section.
References: standardizes the references, according to the Instructions for the authors.
Be careful to specify what the abbreviations refer to the first time they are cited.

Author Response
Reviewer 2
The manuscript describes a study aimed to obtain intragenic octoploid strawberry improved for quality trait under unfavorable climate, through the optimization of in vitro culture protocol and Agrobacterium-mediated intragenic transformation system to produce marker-free transformed plants. The results obtained are interesting but could be described in a more concise and clear manner. Furthermore, the text can be improved, taking care to correct or explain some inconsistencies.
Q1. Extensive English revision by a native speaker is also required, making the manuscript easier to read.
A1: Thank Reviewer for this valuable comment. During revision, we have tried our best to correct the misspellings, divide the very long sentences, specify the certain nouns, and rewrite some sentences with grammatical mistakes. Other than faithfully following the correction suggestions from two Reviewers, we carefully checked our manuscript with the proofreading tool for several times. We are grateful to Reviewer very much. Your suggestions improved our manuscript greatly.
Q2. More suggestions are given to the authors and added directly in the attached text.
A2: Special thanks to the reviewer for those valuable comments. Following Reviewer’s critical annotations, we corrected the main text as well as the supplementary file (Fig. S2) thoroughly. Please see the revised text and sup files. Thank you again.
Q3. Title: “strategy” maybe more suitable than “method”.
A3: Thanks to Reviewer for this suggestion. We have changed this in Title.
Q4. Abstract: The Abstract is too long. According to the Instructions for Authors it should be a total of about 200 words maximum.
A4: Thank Reviewer for the important comment. We have shortened the abstract to provide a pertinent overview the work which presents an objective representation of the background, methods, results and conclusions in 200 words.
Q5. Results: In this Section only the results must be described, also in a robust and clear way. Actually, there is a lot of information that is more suitable for Discussion section.
A5: Thank Reviewer for these valuable suggestions. We have moved the details of optimizing shoot regeneration from Results subsection 2.1 to Materials and Methods 4.1. The presentation of variety characteristics and the selection of DNA components for intragenic strawberry was moved to the first paragraph of Discussion section. Please see Page 15 line 23-30 (from previous version Page 4 Results subsection 2.1); Page 12 line 24-32 (from previous version Page 6 Results subsection 2.2).
Q6. References: standardizes the references, according to the Instructions for the authors.
A6: Thanks to Reviewer very much. We are sorry for some unstandardized presentations of references. Please see the revised text. All references have been checked and corrected. The same response (A10) to Reviewer 3.
Q7. Be careful to specify what the abbreviations refer to the first time they are cited.
A7: Thank Reviewer. Following your comments, we have carefully checked the use of abbreviations. To be specific, ‘TDZ’, ‘IBA’, ‘BA’, ‘MSB5’, and ‘GA’ all were specified in text where they appeared for the first time. Please see Page 4 line11 and line 21, Page 7 line 32, and Page 15 line 18.
Reviewer 3 Report
The authors present an interesting manuscript. The manuscript describes the Agrobacterium-mediated transformation of strawberry leaf explants using a selection system without antibiotics. Compared to previously published data, the authors have achieved the successful generation of putative intragenic plants. Such experiments on fruit crops are still very rare. Another strong point of the manuscript is the development of the ddPCR protocol for determining the copy number of intragenic sequences, which is a non-trivial task, given the high polyploidy and size of the strawberry genome. Therefore, these results are of interest to the “Plants” reader community, especially to plant biotechnology researchers, and results certainly deserve publication.
The limitation of the manuscript is the lack of information regarding the introduction of the vector backbone. It’s well documented that besides the desired sequence, DNA fragments of the plasmid backbone can be detected in some lines of plants produced after marker-free Agrobacterium transformation, therefore these plants are considered not truly cisgenic or intragenic. There is a range of publications including various horticultural crops including potato (doi:10.1038/nbt801; https://doi.org/10.1186/1472-6750-14-50), apple (doi: 10.1111/pbi.12110) and others (see the review doi: 10.1111/pbi.12055). Integration of DNA beyond the borders into the genome of the host plants is reported to occur in 20–75% of transformed plants (doi: 10.1111/pbi.12055). This is especially true for the plants with multiple insertions caused by the spontaneous rearrangement of vector backbone and T-DNA sequence during the introduction. In the recent research on potato among putative intragenic regenerants only the plant with a single copy was truly intragenic, while the plants with several T-DNA insertions have also insertions of PK2oriV and trfA genes outside the T-DNA of the binary vector (DOI 10.1007/s11240-019-01746-9). Taking into account the presence of multiple intragenic insertions in the two independent plants of strawberry (close to integers 6 and 11 copies), reported by the authors, I strongly recommend adding the molecular results (at least PCR) to identify if the produced events are free from construct backbone insertions. Currently, the produced plants are just putatively intragenic, and I even cannot consider them as marker-free ones, since due to a multiple copy insertion, the bacterial selection gene, with a certain degree of probability, could be present in the genome due to a complicated T-DNA integration pattern.
In general an interesting and well-written manuscript, however taking into account the above remark, this manuscript needs a moderate revision.
There are also several issues in the manuscript that need to be addressed. Please improve the English / grammar of the manuscript. There are a few mistakes and other minor errors. Please try to correct them as carefully as possible. Please pay attention to the names of the varieties mentioned, they should be listed as "Hawaii4" (instead of Hawaii4). Please correct similar errors throughout the manuscript (including page 5, figure 1; page 6, text; page 7, figure 2 and text; page 13, text, etc.). There are many errors like mg.L-1 or mg L-1 (page 4; Figure 3, page 16), that must be correct for mg L-1
The introduction is well written, but I highly recommend shortening the text to make it more focused. The part from “Fruit-specific antisense…” till “…of strawberry fruits reversely [17]” (Page 2) is too long. There is no need for an exhaustive overview of strawberry transformation, which is not needed here to understand intragenic experiments.
The results interpretation is mainly appropriate. However, a few technical issues should be addressed for the figures.
Figure 1. The statistical analysis should be added to figure 1a.
Figure 3. The description should be carefully checked and technical errors, like mg.L-1, should be corrected.
Figure 4. Panel d needs to be redone, the PCR is fuzzy, please, add track names or abbreviations, highlight the lanes with the DNA of intragenic plants. It is difficult to understand what is presented, especially given the authors' remark that “The false positive amplification was observed, probably resulted from unspecific amplification or Agrobacterium contamination” (page. 9).
The discussion section is mostly adequate; however, I suggest deleting some of the phrases that are speculative and controversial. In my opinion, the phrase from “The reduced regeneration capability..." till "...could optimize cv. ‘Shanghai Angel’ transformation” (page 13) is too speculative and not supported by manuscript data, so I suggest deleting it to make the text more direct and focused.
Similarly, the phrase “The explant materials should not be sub-cultured more than five times, since a complete loss of shoot regeneration capacity of cv. ‘Shanghai Angel’ has been observed after six-months’ culture for leaf discs from shoots sub-cultured more than eight times (data omitted)” (page 14) also surplus and speculative, should be removed.
After making changes to the manuscript, please add line numbers on each page, this will greatly simplify the review and correction of the text, since it is currently difficult to point out specific mistakes and errors in the text.
Also, please, check the references # 1, 2, 4, 5, 10, 11, 35, 37, 59, 60, 62, 68, 73, 74, 75 for the correctness in accordance with the journal requirement.
In general, to complete the manuscript and validate the data obtained, it is necessary to add the results for the backbone analysis. Finally, given the way the manuscript is currently presented, I believe it needs to be improved as described above to be accepted for publication.
Author Response
Reviewer 3
The authors present an interesting manuscript. The manuscript describes the Agrobacterium-mediated transformation of strawberry leaf explants using a selection system without antibiotics. Compared to previously published data, the authors have achieved the successful generation of putative intragenic plants. Such experiments on fruit crops are still very rare. Another strong point of the manuscript is the development of the ddPCR protocol for determining the copy number of intragenic sequences, which is a non-trivial task, given the high polyploidy and size of the strawberry genome. Therefore, these results are of interest to the “Plants” reader community, especially to plant biotechnology researchers, and results certainly deserve publication.
Q1. The limitation of the manuscript is the lack of information regarding the introduction of the vector backbone. It’s well documented that besides the desired sequence, DNA fragments of the plasmid backbone can be detected in some lines of plants produced after marker-free Agrobacterium transformation, therefore these plants are considered not truly cisgenic or intragenic. There is a range of publications including various horticultural crops including potato (doi:10.1038/nbt801; https://doi.org/10.1186/1472-6750-14-50), apple (doi: 10.1111/pbi.12110) and others (see the review doi: 10.1111/pbi.12055). Integration of DNA beyond the borders into the genome of the host plants is reported to occur in 20–75% of transformed plants (doi: 10.1111/pbi.12055). This is especially true for the plants with multiple insertions caused by the spontaneous rearrangement of vector backbone and T-DNA sequence during the introduction. In the recent research on potato among putative intragenic regenerants only the plant with a single copy was truly intragenic, while the plants with several T-DNA insertions have also insertions of PK2oriV and trfA genes outside the T-DNA of the binary vector (DOI 10.1007/s11240-019-01746-9). Taking into account the presence of multiple intragenic insertions in the two independent plants of strawberry (close to integers 6 and 11 copies), reported by the authors, I strongly recommend adding the molecular results (at least PCR) to identify if the produced events are free from construct backbone insertions. Currently, the produced plants are just putatively intragenic, and I even cannot consider them as marker-free ones, since due to a multiple copy insertion, the bacterial selection gene, with a certain degree of probability, could be present in the genome due to a complicated T-DNA integration pattern.
A1: Thank Reviewer for your valuable comment. This comment really improved our work very much. Following your suggestion, we designed primers matching to the antibiotic marker gene beyond T-DNA region of the vector we constructed. PCR analysis of the two complete transformants revealed the integration of vector backbone in these transformants, indicating that they are not true intragenic strawberry. Accordingly, we have corrected our manuscript: the PCR analysis for vector backbone was added to Figure 4 as part (4e); the result, discussion, the introduction, as well as the abstract and manuscript title all have been modified, thus to be consistent with the experiment results. We believe it is still hopeful to reveal true intragenic strawberry if we develop more transformants in the future. Thank you so much for help us expanding knowledge and correcting errors.
Q2. In general an interesting and well-written manuscript, however taking into account the above remark, this manuscript needs a moderate revision.
There are also several issues in the manuscript that need to be addressed. Please improve the English / grammar of the manuscript. There are a few mistakes and other minor errors. Please try to correct them as carefully as possible. Please pay attention to the names of the varieties mentioned, they should be listed as "Hawaii4" (instead of Hawaii4). Please correct similar errors throughout the manuscript (including page 5, figure 1; page 6, text; page 7, figure 2 and text; page 13, text, etc.). There are many errors like mg.L-1 or mg L-1 (page 4; Figure 3, page 16), that must be correct for mg L-1
A2: Thank Reviewer for the critical comment. We are sorry for the errors in variety names and units for chemical concentration. Following Reviewer’s suggestion, we have checked the main text thoroughly, and carefully corrected these errors. Please see the revised version. Thank you again.
Q3. The introduction is well written, but I highly recommend shortening the text to make it more focused. The part from “Fruit-specific antisense…” till “…of strawberry fruits reversely [17]” (Page 2) is too long. There is no need for an exhaustive overview of strawberry transformation, which is not needed here to understand intragenic experiments.
A3: Thank Reviewer for the critical comment. During revision, we deleted four references of basic researches concerning gene functional studies (original [13-16]). Simultaneously, the missed work of Schaart et al. (2011) was included in the same paragraph. Numbering references was altered accordingly. Please see Page 2 line 27-34.
Q4. The results interpretation is mainly appropriate. However, a few technical issues should be addressed for the figures.
Figure 1. The statistical analysis should be added to figure 1a.
A4: Thanks to Reviewer. We are sorry for missing the statistical analysis. Figure 1a has been improved with statistical analysis added. Please see Page 5.
Q5. Figure 3. The description should be carefully checked and technical errors, like mg.L-1, should be corrected.
A5: Thanks to Reviewer. We have carefully checked the legend for Figure 3 and corrected technical errors in Figure 3. Please see the highlighted legend after revision (Page 8 line 16-24).
Q6. Figure 4. Panel d needs to be redone, the PCR is fuzzy, please, add track names or abbreviations, highlight the lanes with the DNA of intragenic plants. It is difficult to understand what is presented, especially given the authors' remark that “The false positive amplification was observed, probably resulted from unspecific amplification or Agrobacterium contamination” (page. 9).
A6: Thank Reviewer for this important comment. We are sorry for not providing the right version of Figure 4d in the previous submission. We have redone the direct PCR analysis for regenerated shoots after transformation and provided a new Figure 4d. In addition, it’s true that weak false positive amplification was randomly observed in direct PCR for some shoots which were later confirmed to be untransformed when their genomic DNAs were purified and examined in a second PCR. We provided a clarification in this section. Please see Page 9 line 32-40.
Q7. The discussion section is mostly adequate; however, I suggest deleting some of the phrases that are speculative and controversial. In my opinion, the phrase from “The reduced regeneration capability..." till "...could optimize cv. ‘Shanghai Angel’ transformation” (page 13) is too speculative and not supported by manuscript data, so I suggest deleting it to make the text more direct and focused.
A7: Thank Reviewer for this critical comment. Following your suggestion, we deleted these speculative phrases in discussion followed with two references (previous [66-67]). Please see the revised version (Page 13, line 38).
Q8. Similarly, the phrase “The explant materials should not be sub-cultured more than five times, since a complete loss of shoot regeneration capacity of cv. ‘Shanghai Angel’ has been observed after six-months’ culture for leaf discs from shoots sub-cultured more than eight times (data omitted)” (page 14) also surplus and speculative, should be removed.
A8: Changed as suggested. Thanks a lot. Please see Page 14 line 12.
Q9. After making changes to the manuscript, please add line numbers on each page, this will greatly simplify the review and correction of the text, since it is currently difficult to point out specific mistakes and errors in the text.
A9: Thanks for the important suggestion. We are very sorry for missing the line numbers. Please see the revised manuscript with line numbers.
Q10. Also, please, check the references # 1, 2, 4, 5, 10, 11, 35, 37, 59, 60, 62, 68, 73, 74, 75 for the correctness in accordance with the journal requirement.
A10: Thanks to Reviewer for the valuable and critical comment. During revision, we have standardized all the references according to the instructions for authors. Please see the highlighted lists in revised version after correction.
Q11. In general, to complete the manuscript and validate the data obtained, it is necessary to add the results for the backbone analysis. Finally, given the way the manuscript is currently presented, I believe it needs to be improved as described above to be accepted for publication.
A11: The same response to question 1 (Q1-A1). Thank Reviewer very much.
Round 2
Reviewer 1 Report
The authors were able to regenerate 4 transgenic plants, of which 2 were with incomplete T-DNA integration and 2 were multicopy. From this point of view, the title of the article still does not describe the obtained results and still it is necessary to improve the protocol to obtain transgenic plants with sufficient expression of the introduced genes. I appreciate the large number of experiments and the results achieved, but they are only at the beginning. Regeneration of strawberries without selection pressure is interesting, but as it turned out, it also brings problems. The problems should be stated correctly, and interesting results should be emphasized (eg ddPCR protocol for copy number determination).
Others:
“…Actually, as shown in Fig. 1c, most leaf explants 9 of cv. ”Shanghai Angel“ from micropropagation regenerated via caulogenesis on wounding sites, which would reduce the frequency of chimerism during transformation…”
On what basis do you think that injured cells are more likely to produce callus from only 1 cell? Add reference for this or please reconsider this sentence.
The Figure and legend are still illegible to me. Which means 2x 13-7 and 20-7?
„As shown in Figure 4e, the two independent lines tested were true transformants, but not free of vector backbone“. I'm not sure if there were still some bacterial cells on the plant material that was analyzed (KmR). PCR for the presence of viral genes would help.
Author Response
Reviewer 1
The authors were able to regenerate 4 transgenic plants, of which 2 were with incomplete T-DNA integration and 2 were multicopy. From this point of view, the title of the article still does not describe the obtained results and still it is necessary to improve the protocol to obtain transgenic plants with sufficient expression of the introduced genes. I appreciate the large number of experiments and the results achieved, but they are only at the beginning. Regeneration of strawberries without selection pressure is interesting, but as it turned out, it also brings problems. The problems should be stated correctly, and interesting results should be emphasized (eg ddPCR protocol for copy number determination).
R: Thank Reviewer for your valuable comments. We have corrected the Abstract, the Results, and Discussion sections based on what we achieved and revealed in this work. Please see the revised main text.
Others:
“…Actually, as shown in Fig. 1c, most leaf explants of cv. ‘Shanghai Angel’ from micropropagation regenerated via caulogenesis on wounding sites, which would reduce the frequency of chimerism during transformation…”
On what basis do you think that injured cells are more likely to produce callus from only 1 cell? Add reference for this or please reconsider this sentence.
R: In current work, leaf explants were pre-cultured before transformation for a duration of 2-4 weeks, which enables the preparation of regeneration status in explants. The regeneration pathways affect the frequency of chimerism, and it is believed that a higher chimerism will occur in organogenesis than in caulogenesis, which we provided reference [64] in the same paragraph of Discussion section (Page 14, line 10). Thank you for your understanding.
The Figure and legend are still illegible to me. Which means 2x 13-7 and 20-7?
R: The DNA samples from independent lines #13-1 and #20-7 were tested in duplicate repeats.
“As shown in Figure 4e, the two independent lines tested were true transformants, but not free of vector backbone„. I'm not sure if there were still some bacterial cells on the plant material that was analyzed (KnR). PCR for the presence of viral genes would help.
R: Templates for PCR analysis displayed in Figure 4e were purified genomic DNA samples. As we showed in Material and Method section, the genomic DNAs were purified from candidate transformants with a Super Plant Genomic DNA kit for polysaccharides and polyphenolics-rich samples (TianGen, Beijing, Cat. No. DP360) (Page 18, Line 2-4). Electrophoresis on 1.0% agarose gel integrated with quantification using Qubit 4.0 Fluorometer jointly suggested that the DNA samples were clean and free of bacterial contaminations.
Reviewer 2 Report
The manuscript has greatly improved but still some corrections are needed. Some comments and suggestions are added directly into the text

Author Response
Reviewer 2:
The manuscript has greatly improved but still some corrections are needed. Some comments and suggestions are added directly into the text.
R: Thank Reviewer for your valuable comments and suggestions. We have performed correction following the guidance. Please see the tracked version of our revised manuscript. We do appreciate your correction and help very much.
Reviewer 3 Report
The manuscript has improved but some issues in the manuscript need to be addressed.
In the first regeneration experiment that involved aseptic leaf material, the regeneration rate was raised 55% with the best TDZ concentration (Figure 1a, page 4 line 12). In the next experiment (aseptic compared to greenhouse), the regeneration rate decreased by more than two times to 15-20% in ascetic explants at the same medium. What happened? If the authors claim that a certain combination can statistically improve regeneration using in vitro material (Fig. 1a) (in the first version the statistical analysis was missing, now is present), why in the new experiment (Fig. 1b) a similar level of regeneration was not observed for the same material? This requires clarifying or figure 1a should be removed, at least. To be honestly the figure 1C is not providing clear information proving differences between aseptic and greenhouse leaves. Taking into account the inexplicable observations in Figure 1a and 1b, I advise moving Figure 1 to supplementary material. There is a more informative Figure 3.
Supplementary figure S1 indicated on page 7 (line 12) is not provided in the supplementary material.
There are some technical issues in the text:
page 2 line 44 replace Calypco to ‘Calypco’
page 3 line 41 replace genetic to genetically
page 9 line 37 replace ass to as
page 10, lines 19, 27, 33 replace “Shanghai Angel” to ‘Shanghai Angel’
page 13 lines 18, 33 replace “Shanghai Angel” to ‘Shanghai Angel’, line 22 replace Rapella to ‘Rapella’ and Redcoat to ‘Redcoat’; line 24 replace Firework to ‘Firework’; line 26 replace Chandler to ‘Chandler’; line 29 replace Laboratory Festival #9 to ‘Laboratory Festival #9’.
Page 14 lines 5, 10 replace “Shanghai Angel” to ‘Shanghai Angel’, line 21 replace Fielder to ‘Fielder’; line 26 replace Calypso to ‘Calypso’
Page 15 lines 3, 6, 7, 23 replace “Shanghai Angel” to ‘Shanghai Angel’, line 4 replace Benihoppe x Akihime to ‘Benihoppe’ x ‘Akihime’. Line 37 replace “Hawaii4” to ‘Hawaii4’
Page 16 line 9 replace “Hawaii4” to ‘Hawaii4’; line 23 replace “Shanghai Angel” to ‘Shanghai Angel’
Knowing that the produced plants were not truly intragenic and the bacterial Kanamycin resistance gene was also present in the plant genome, the section ”2.3 Developing marker-free intragenic strawberry plants via Agrobacterium-mediated transformation without selection” needs the modification like “Developing putative intragenic strawberry plants via…”
Technical issues in figures:
Figure 1
replace “Shanghai Angel” to ‘Shanghai Angel’
Figure 2
replace”Hawaii4” to ‘Hawaii4’(line 40, 41, 42, 43)
Figure 4
replace “Shanghai Angel” to ‘Shanghai Angel’
Figure 5
replace “Shanghai Angel” to ‘Shanghai Angel’
References:
Among the four references [16-19] used in the Introduction section (page 2 lines 39-40), the absence of bacterial marker genes from the vector backbone was not demonstrated in the manuscript [17], so it’s unknown if the produced potato plants were truly marker-free; this reference could be replaced with another one (Miroshnichenko, D.; Timerbaev, V.; Okuneva, A.; Klementyeva, A.; Sidorova, T.; Pushin, A.; Dolgov, S. Enhancement of resistance to PVY in intragenic marker-free potato plants by RNAi-mediated silencing of eIF4E translation initiation factors. Plant .Cell Tiss. Organ. Cult. 2020, 140, 691–705.)
A lot of references (##1, 9, 14, 34, 41, and so on) still have technical errors in the names of journals, please correct.
Author Response
Reviewer 3:
The manuscript has improved but some issues in the manuscript need to be addressed.
In the first regeneration experiment that involved aseptic leaf material, the regeneration rate was raised 55% with the best TDZ concentration (Figure 1a, page 4 line 12). In the next experiment (aseptic compared to greenhouse), the regeneration rate decreased by more than two times to 15-20% in ascetic explants at the same medium. What happened? If the authors claim that a certain combination can statistically improve regeneration using in vitro material (Fig. 1a) (in the first version the statistical analysis was missing, now is present), why in the new experiment (Fig. 1b) a similar level of regeneration was not observed for the same material? This requires clarifying or figure 1a should be removed, at least. To be honestly the figure 1C is not providing clear information proving differences between aseptic and greenhouse leaves. Taking into account the inexplicable observations in Figure 1a and 1b, I advise moving Figure 1 to supplementary material. There is a more informative Figure 3.
R: The regeneration efficiency in Fig. 1a was obtained after 60 days’ in vitro culture. In the following comparative study on different explants (Fig. 1b), the regeneration efficiency was evaluated after 42 days (six weeks) in vitro culture. Please see the legend of Figure 1. Concerning Fig. 1c, we think it provided the evidences for direct organogenesis on greenhouse-derived explants as early as three weeks post in vitro culture (as indicated by the red arrows), which is vital for the understanding of our selection of explants in genetic transformation, and the resulting chimeras we obtained. Thank you for your understanding.
Supplementary figure S1 indicated on page 7 (line 12) is not provided in the supplementary material.
R: The vector backbone of pMDC162 beyond T-DNA region contains three expression cassettes, namely, the aminoglycoside phosphotransferase conferring resistance to kanamycin (KanR), the stability protein from Pseudomonas plasmid pVS1 (pVS1 StaA), and the replication protein from Pseudomonas plasmid pVS1 (pVS1 RepA). More information about this vector could be found in previous publication: Curtis MD, Grossniklaus U. A gateway cloning vector set for high-throughput functional analysis of genes in planta. Plant Physiol. 2003;133(2):462-469. doi:10.1104/pp.103.027979.
Since we are preparing for a new work and hope to provide a comparative study of transformation efficiency with the vector used in this study and two additional vectors, we hope to display their circular maps in that near future work. Thank you for your understanding.
There are some technical issues in the text:
page 2 line 44 replace Calypco to ‘Calypco’
page 3 line 41 replace genetic to genetically
page 9 line 37 replace ass to as
page 10, lines 19, 27, 33 replace “Shanghai Angel” to ‘Shanghai Angel’
page 13 lines 18, 33 replace “Shanghai Angel” to ‘Shanghai Angel’, line 22 replace Rapella to ‘Rapella’ and Redcoat to ‘Redcoat’; line 24 replace Firework to ‘Firework’; line 26 replace Chandler to ‘Chandler’; line 29 replace Laboratory Festival #9 to ‘Laboratory Festival #9’.
Page 14 lines 5, 10 replace “Shanghai Angel” to ‘Shanghai Angel’, line 21 replace Fielder to ‘Fielder’; line 26 replace Calypso to ‘Calypso’
Page 15 lines 3, 6, 7, 23 replace “Shanghai Angel” to ‘Shanghai Angel’, line 4 replace Benihoppe x Akihime to ‘Benihoppe’ x ‘Akihime’. Line 37 replace “Hawaii4” to ‘Hawaii4’
Page 16 line 9 replace “Hawaii4” to ‘Hawaii4’; line 23 replace “Shanghai Angel” to ‘Shanghai Angel’
Knowing that the produced plants were not truly intragenic and the bacterial Kanamycin resistance gene was also present in the plant genome, the section ”2.3 Developing marker-free intragenic strawberry plants via Agrobacterium-mediated transformation without selection” needs the modification like “Developing putative intragenic strawberry plants via…”
Technical issues in figures:
Figure 1: replace “Shanghai Angel” to ‘Shanghai Angel’
Figure 2: replace ”Hawaii4” to ‘Hawaii4’(line 40, 41, 42, 43)
Figure 4: replace “Shanghai Angel” to ‘Shanghai Angel’
Figure 5: replace “Shanghai Angel” to ‘Shanghai Angel’
R: All have been corrected as Reviewer suggested. Thank you.
References:
Among the four references [16-19] used in the Introduction section (page 2 lines 39-40), the absence of bacterial marker genes from the vector backbone was not demonstrated in the manuscript [17], so it’s unknown if the produced potato plants were truly marker-free; this reference could be replaced with another one (Miroshnichenko, D.; Timerbaev, V.; Okuneva, A.; Klementyeva, A.; Sidorova, T.; Pushin, A.; Dolgov, S. Enhancement of resistance to PVY in intragenic marker-free potato plants by RNAi-mediated silencing of eIF4E translation initiation factors. Plant .Cell Tiss. Organ. Cult. 2020, 140, 691–705.)
R: Thank Reviewer very much. We performed the correction.
A lot of references (##1, 9, 14, 34, 41, and so on) still have technical errors in the names of journals, please correct.
R: Corrected. Thanks a lot.